# ROS and miRNA Dysregulation in Ovarian Cancer Development, Angiogenesis and Therapeutic Resistance

**DOI:** 10.3390/ijms23126702

**Published:** 2022-06-16

**Authors:** David C. Stieg, Yifang Wang, Ling-Zhi Liu, Bing-Hua Jiang

**Affiliations:** 1Department of Medical Oncology, Sidney Kimmel Cancer Center, Thomas Jefferson University, Philadelphia, PA 19107, USA; dcstieg@gmail.com (D.C.S.); ling-zhi.liu@jefferson.edu (L.-Z.L.); 2Department of Pathology, Anatomy & Cell Biology, Sidney Kimmel Cancer Center, Thomas Jefferson University, Philadelphia, PA 19107, USA; yifang.wang@students.jefferson.edu

**Keywords:** ovarian cancer, miRNA dysregulation, ROS, NOX4, HIF1-α, VEGF, angiogenesis, HER3, therapeutic resistance

## Abstract

The diverse repertoires of cellular mechanisms that progress certain cancer types are being uncovered by recent research and leading to more effective treatment options. Ovarian cancer (OC) is among the most difficult cancers to treat. OC has limited treatment options, especially for patients diagnosed with late-stage OC. The dysregulation of miRNAs in OC plays a significant role in tumorigenesis through the alteration of a multitude of molecular processes. The development of OC can also be due to the utilization of endogenously derived reactive oxygen species (ROS) by activating signaling pathways such as PI3K/AKT and MAPK. Both miRNAs and ROS are involved in regulating OC angiogenesis through mediating multiple angiogenic factors such as hypoxia-induced factor (HIF-1) and vascular endothelial growth factor (VEGF). The NAPDH oxidase subunit NOX4 plays an important role in inducing endogenous ROS production in OC. This review will discuss several important miRNAs, NOX4, and ROS, which contribute to therapeutic resistance in OC, highlighting the effective therapeutic potential of OC through these mechanisms.

## 1. Ovarian Cancer

Known as the silent killer, ovarian cancer (OC) has the lowest survival rate and the worst prognosis among all gynecologic malignancies in the US; and is the eighth most common cancer in women worldwide [1,2]. In 2022, the American Cancer Society estimates about 21,000 new cases of OC will be diagnosed, and approximately 14,000 women will die from this type of cancer. The overall 5-year survival rate is only 48% due to OC’s ambiguous symptoms and inadequate screening capabilities at the early stages of the disease. Due to late detection, about 60% of new cases are diagnosed when the disease has already progressed to the advanced stage [2]. OC is a heterogeneous disease with several subtypes that differ in their gene expression, tumor origin, pathway alterations, and pathogenesis. The majority of OC originates from three main cell types: epithelial cells (90%), stromal cells (7%), and germ cells (3%) [1,3,4]. In general, epithelial OC can be further divided into five histotypes: high-grade serous (HGSOC; 70%), endometrioid (ENOC; 10%), clear cell (CCOC; 10%), mucinous (MOC; 5%), and low-grade serous (LGSOC; less than 5%) OC [4]. In addition, another classification system was introduced a decade ago that divided OC into type I and II tumors. Type I tumors are low-grade neoplasms, including mucinous carcinomas, endometrioid carcinomas, malignant Brenner tumors, and clear cell carcinomas. Type I tumors are typically characterized by mutations in BRAF, KRAS, and PTEN with DNA instability. Type II tumors are high-grade serous carcinoma, carcinosarcoma, and undifferentiated carcinoma, which are frequently observed with mutations in p53, BRCA1/2, HER-2/HER-3 overexpression, and p16 inactivation [5,6,7,8]. Depending on the specific subtype and histopathology, OC treatment involves a combination of surgery and chemotherapy. For patients with advanced-stage tumors, debulking surgery is recommended; however, large tumors or residual tumors may show negative side effects leading to blockage of the perfusion area and the possibility of developing drug resistance [1,9]. Platinum-based chemotherapy is the standard line of treatment for OC, either in conjunction with or following surgery [10,11,12]. The combination of paclitaxel/carboplatin has been recognized as the standard postoperative chemotherapy for many years [13]. In recent years, PARP inhibitors have been incorporated into clinical treatment as a recommended maintenance drug [14]. However, due to the aggressive growth rates and the propensity of advanced tumors to evade treatment, there are critical limitations to the current lines of therapy. A better understanding of the molecular biology of OC is allowing more research efforts to establish new effective treatment options for advanced-stage tumors.

## 2. ROS

Reactive oxygen species (ROS) have remained a highly relevant topic over the last few decades due to their expansive effects on normal cellular function. Oxidative stress is generated through the accumulation of ROS, either through exogenous exposure or endogenous production. ROS are oxygen ions with unpaired electrons (singlet oxygen ^1^O_2_, superoxide O_2_·^−^) or oxygen-containing molecules, such as hydroxyl radicals (OH·^−^), hydrogen peroxide (H_2_O_2_), nitric oxide (NO), and nitrogen dioxide (NO_2_) [15]. Superoxide radicals are converted into H_2_O_2_ by the enzyme superoxide dismutase (SOD). However, superoxide can also react with nitric oxide to produce peroxynitrite (ONOO^−^), a strong oxidizer with damaging cellular effects [16]. The accumulation of H_2_O_2_ has detrimental effects on nuclear and mitochondrial DNA, which may lead to genetic instability to drive cancer progression with increased expression of oncogenes and decreased expression of tumor suppressors [17,18]. Several enzymes work in conjunction to convert H_2_O_2_ into the water, including catalase, glutathione peroxidases 1 and 4, and peroxiredoxins 3 and 5 [19,20,21,22]. Furthermore, H_2_O_2_ can also participate in the Fenton reaction, in which free iron Fe(II) reacts with H_2_O_2_, generating highly reactive hydroxyl radicals (·OH)(shown below). The production of hydroxyl radicals (·OH) by the Haber–Weiss reaction (shown below) further perpetuates the damaging effects of the accumulation of ROS.


**Fenton Reaction:**
Fe(II) + H_2_O_2_ ⟷ Fe(III) + ·OH + OH^−^




**Haber–Weiss Reaction:**
O_2_·^−^ + H_2_O_2_ ⟷ OH + OH^−^ + O_2_


Original studies implicated the mitochondria as primary endogenous sources of superoxide through the process of cellular respiration, a process dependent on the availability of O_2_ [23,24,25]. Based on this view, the production of ROS was thought to be a harmful by-product of intracellular metabolism. Then a family of transmembrane enzymes known as NADPH oxidase (NOX) proteins was identified, whose primary function was the production of endogenous ROS. NOX2, the first NOX protein discovered, was the primary producer of endogenous ROS in leukocytes to generate an oxidative burst, an essential process for the neutralization of pathogens [26,27,28,29]. The characterization of a disease called chronic granulomatous disease (CGD) caused by a mutation in the phagocytic NOX gene provided insight into the emerging role of endogenous ROS production on cellular functionality [30,31]. Subsequent work demonstrated a pivotal role of NOX proteins in mammalian cell transformation through the production of superoxide radicals and H_2_O_2_ [32,33]. Our group demonstrated that the accumulation of ROS in OC cells was attributed to H_2_O_2_ increased levels induced by NOX4 [34], identifying an endogenous mechanism for the overproduction of ROS and alteration of intracellular signaling in OC tumor development.

Under normal cellular conditions, low levels of endogenous ROS activate several signaling pathways involved in cell proliferation. However, the accumulation of ROS causes extensive damage to DNA, RNA, proteins, and lipids, thus causing a significant hindrance to normal cellular functions and contributing to the development of multiple human pathologies [35,36,37,38]. The damage can induce cell death pathways or trigger the mutation of DNA, as commonly found in cancer [39,40]. In addition to the endogenous production of ROS and oxidative stress, external or environmental exposure to ROS can have detrimental effects on mammals [41]. For instance, many chemotherapeutic agents induce oxidative stress as a means of inducing cellular damage and cell death pathways [42]. However, as demonstrated by more recent findings, ROS play an important role in the progression and advancement of human diseases. The counterweight for endogenous ROS is the genetically programmed redox system. This includes groups of genes coding for antioxidant proteins such as superoxide dismutase (SOD), catalase, and the glutathione system, which neutralize the ROS produced in cells [43,44,45]. The failure to neutralize endogenous ROS leads to a build-up of harmful oxygen species and, consequently, oxidative stress. In normal cells, oxidative stress leads to deleterious cellular effects, such as protein, lipid, and DNA damage, organelle dysfunction, and cell cycle arrest [46]. Higher levels of oxidative stress cause the activation of cell death pathways such as apoptosis and necrosis [46], which may be mitigated in cancer cells by an increase in antioxidant production. The upregulation of nuclear factor erythroid 2-related factor 2 (NRF2), a master transcriptional regulator of antioxidant genes, contributes to the neutralization of endogenous ROS in OC cells [47,48,49], making NRF2 a viable target for chemotherapeutic treatment in certain cases of OC. In addition, the genetic mutation of cellular pathways that induce cell death mechanisms in response to increased oxidative stress allows cancer cells to evade the activation of cell death pathways [50], thus providing cancer cells the ability to continue continuous proliferation in the presence of adverse cellular conditions, such as oxidative stress.

## 3. ROS in the Development of Ovarian Cancer

There is an established link between an increase in ROS production and cancer development in humans [51]. As secondary cellular signaling molecules, ROS are involved in the activation of several signaling pathways involved in cell proliferation and growth. Consequently, these pathways are constitutively activated in cancer cells with increased ROS levels that contribute to tumorigenesis [51]. For example, endogenously derived ROS activate the ERK1/2 MAPK signaling pathway and the AKT signaling pathway in OC, both of which promote cell proliferation [52,53]. The increased ROS generation also contributes to a genetic mutation in cancer cells, further contributing to cell transformation [54,55]. As opposed to the traditional view of ROS generation in cancer as a harmful secondary by-product, the increasing knowledge of cancer cell metabolism and signal transduction is exposing ROS as a positive contributing factor in tumorigenesis and cancer development. The increased metabolic activity of cancer cells was originally thought to be responsible for the accumulation of ROS as a byproduct of increased glycolytic metabolism and mitochondrial respiration [56]. However, the discovery of the role of NOX proteins in endogenous ROS production revealed a more important role for ROS production in non-phagocytic cells, particularly in cancer [57,58,59]. The endogenous production of ROS by NOX1 was found to be responsible for increased viability and proliferation in colon cancer [60,61]. Similarly, the role of NOX2-mediated ROS production was discovered to be critical for cell viability and proliferation in breast, colorectal, myelomonocytic leukemia, gastric, and prostate cancers [62,63,64,65,66,67]. NOX4 overexpression contributed to an oncogenic proliferation in renal cell carcinoma, melanoma, glioblastoma, ovarian, prostate, and lung cancers [34,68,69,70,71,72]. In OC cell lines, there is a significant increase in ROS production, which contributes to tumorigenesis [34]. The increase in ROS is a result of NADPH oxidase activity and mitochondrial metabolism, as this increase is diminished by NADPH oxidase and mitochondrial complex I inhibitors [34]. Moreover, the increased levels of ROS result from the upregulation of the NADPH oxidase subunit NOX4, which serves as the main contributor to ROS production in OC cells to promote tumor growth and angiogenesis [34]. Furthermore, the activation of NOX4 is positively correlated with TGF-β1 and NF-κB activity, which is suppressed by their inhibitors [34]. This system demonstrates that endogenous NOX4-derived ROS are a driving force in OC development. Moreover, NOX4 is a potential target for the therapeutic resistance of OC which is dependent on ROS production for an increase in oncogenic signaling.

## 4. miRNA Dysregulation in Cancer

The progression of cancer is often associated with dysregulation of non-coding RNAs, including microRNAs (miRNAs) [73,74]. miRNAs are 18–25 nucleotide long, non-coding single-stranded RNA molecules that regulate the expression of messenger RNA (mRNA) [75]. The discovery of miRNA in 1993 by Ambros and colleagues in the nematode *C. Elegans* revealed the critical role of miRNAs in the post-transcriptional regulation of mRNA [76,77]. In these studies, the miRNA lin-4 was found to regulate the expression of the critical developmental transcription factor, lin-14 [76,77]. The primary transcripts of miRNA (pri-miRNA) are modified within the nucleus by the RNase III DROSHA and its cofactor DGCR8 before being exported to the cytoplasm as pre-miRNA [78,79]. Mature miRNA molecules are the result of the cleavage of pre-miRNA at the terminal loop by the RNase III endonuclease, DICER [80,81]. The regulation of miRNA processing can have expansive effects on cellular processes, as demonstrated by the gain-of-function mutation of DICER as a contributory factor in cancer development [82]. As transcriptional regulatory molecules, miRNAs typically recognize and bind the 3′-UTR of target mRNAs to repress expression or induce degradation [83]. The activation of genes by miRNAs occurs through association with the promoter region and upstream regulatory regions of target genes [84]. The search for the role of miRNA in humans yielded a plethora of data that are still accumulating, particularly the dysregulation of miRNAs in oncogenesis. The original studies identifying the role of miRNAs in human oncogenesis demonstrated the effect of miR-15a/16a repression on promoting the oncogenic protein Bcl-2 in chronic lymphocytic leukemia [85]. Most human miRNAs function as tumor suppressors by directly targeting and inhibiting oncogenes, such as RAS and MYC. For instance, the downregulation of Let-7 family of miRNAs, which target KRAS and C-MYC, is found in OC, which induces tumor growth and development [86,87]. However, some miRNAs function as oncogenes by directly targeting and inhibiting tumor suppressors such as p53 [88,89]. For example, miR-25 and miR-30d target p53 for degradation and contribute to colon cancer development; the downregulation of both miR-25 and miR-30d led to an increase in p53 protein expression and increased apoptosis in multiple cancer types [90]. Many miRNAs are dysregulated in multiple cancers, including the upregulation of miR-155 in lymphomas and colorectal cancers [91,92], indicating a commonality in the mode of miRNA dysregulation in multiple cancer/tissue types. The molecular effects of miRNA dysregulation include feedback mechanisms, such as the miR-17-92 cluster/E2F family/c-MYC loop. In this feedback mechanism, miR-17-92 is activated by c-MYC and inhibits E2F family protein translation [93,94]. The E2F family of proteins (E2F1, E2F2, E2F3) are critical cell-cycle regulated inducers of proliferation, therefore proper regulation of these proteins is necessary under normal conditions [95]. Further investigation revealed that c-MYC activation of E2F family proteins activates miR-17-92, leading to a feedback loop to tightly control the expression of E2F proteins in healthy cells [96,97]. However, in cancer cells the amplification and overexpression of miR-17-92 disrupts this feedback loop and contributes to high cell proliferation and tumorigenesis [98]. In another example, miR-221/222 upregulation in cancer cells contributes to oncogenesis through the inhibition of cell cycle regulating protein p27 [99,100,101]. The dysregulation of particular miRNAs can differ between subtypes of OC. For instance, the overexpression of miR-483 occurs in serous epithelial ovarian cancer (EOC), but does not occur in non-serous EOC [102]. As demonstrated in voluminous publications, miRNA dysregulation affects a variety of cellular processes that contribute to oncogenesis in a wide variety of cancers. The complex role of miRNAs in cancer development highlights the potential for therapies targeting specific miRNAs that are dysregulated in different cancers.

## 5. ROS and miRNA Dysregulation in Angiogenesis and Ovarian Cancer Development

The development of tumors involves a wide variety of cellular processes. In this regard, ROS contribute to critical cellular processes that occur within tumors, including angiogenesis and micro-RNA (miRNA) dysregulation. Angiogenesis is the creation of new blood vessels within existing vasculature, which is essential for processes such as embryogenesis, tissue repair, and organ regeneration [103]. Unsurprisingly, angiogenesis plays a pivotal role in cancer development through the establishment of nutrients and blood supply to newly formed tumors [104]. A significant contribution to angiogenic signaling is made by vascular endothelial growth factor (VEGF), which is highly upregulated in developing embryonic cells and tumor cells [105,106,107]. The limited oxygen availability in tumors often leads to hypoxic conditions, in which signaling pathways are activated to initiate tumor growth and angiogenesis [108]. The hypoxia-inducible factor 1 alpha (HIF-1α) plays a vital role in the hypoxic response in tumor cells, partially through the upregulation of VEGF [109,110,111]. The dysregulation of HIF1-α occurs in a wide variety of cancers which contributes to tumorigenesis [112]. The upregulation of HIF-1α and VEGF are positively correlated with NOX4-derived ROS production in OC cells and promotes angiogenesis and tumor growth [34]. In turn, HIF-1α induces the expression of VEGF; and promotes the production of NOX4 through an alternative splicing mechanism [34,113]. This positive feedback system demonstrates the capacity of OC cells to utilize the overproduction of NOX4-derived ROS to activate HIF-1α and VEGF and promote angiogenesis and tumor growth.

This regulation system is complicated more by the dysregulation of miRNAs by increased intracellular ROS, which contributes to OC tumorigenesis. For instance, miR-199a and miR-125b downregulate the expression of the oncogenic proteins HER2 and HER3 under normal cellular conditions [114]. However, increased ROS in OC cells results in the downregulation of miR-199a and miR-125b through DNA hypermethylation, thereby increasing the expression of HER2/3 and contributing to tumorigenesis [114]. More importantly, this demonstrates the role of epigenetic regulation of miRNA expression. Another example of epigenetic regulation of miRNA expression is the increased acetylation of miR-466-5p in response to increased ROS, thereby inducing the expression of miR-466-5p, which then activates pro-apoptotic genes [115,116]. The expression of some miRNAs is regulated by ROS-stress-responsive transcription factors, which contribute to molecular signaling cascades. The tumor suppressor p53 is induced in response to ROS and subsequently activates the miR-200 family of miRNAs [117]. The miR-200 family members have been implicated as tumor suppressors, and overexpression of these miRNAs inhibits tumor development in OC [118,119]. However, two members of this family of miRNAs, miR-141 and miR-200a, directly target p38α in response to increased levels of ROS, resulting in evasion of apoptosis induction and upregulation in antioxidant production [120,121], thus demonstrating the complexity of molecular roles of a single family of miRNAs in OC progression. As contributing factors to the pathway described above, miR-21 and miR-27a induce angiogenesis in OC through the upregulation of HIF1-α and VEGF, respectively [122,123]. Additional pathways affected by miRNA dysregulation also contribute to angiogenesis in OC. For instance, miR-141 is upregulated in OC to induce the expression of VEGFR2, resulting in an increase in angiogenesis [124]. By a differing mechanism, the upregulation of miR-205 in OC results in an increase in angiogenesis through the downregulation of tumor suppressor PTEN and an increase in AKT signaling [125]. Similarly, miR-204 upregulation in OC contributes to angiogenesis through the downregulation of anti-angiogenic protein THBS1 [126,127]. The miRNAs listed in Table 1 are upregulated in OC cells and contribute positively to tumor growth, development, angiogenesis, and therapeutic resistance. However, there are substantial data demonstrating a tumor suppressor role for various miRNAs in OC, whose downregulation results in tumorigenesis, angiogenesis, and treatment resistance (Table 2) [128]. For instance, *miR-145* acts as a tumor suppressor, and the downregulation of miR-145 in OC contributes to angiogenesis through the upregulation of HIF-1α and VEGF [129]. The overall role of the miRNAs in OC development described here is shown in Figure 1. Another important avenue of miRNA research involves the dysregulation of specific circulatory miRNAs, which has the inherent propensity to impact a multitude of tissue types. Recent work has shown that the expression levels of miR-200b, miR-200c, miR-141, and miR-1274A in OC patients’ circulatory systems are negatively correlated with survival [130]. Therefore, the roles of miRNA dysregulation in OC angiogenesis and development remain to be fully understood which warrants further investigation to provide therapeutic options and/or targets in the future.

## 6. Potential Mechanism of ROS in Therapeutic Resistance in Ovarian Cancer

The two major obstacles facing efficient treatment of OC are late detection/diagnosis and acquired therapeutic resistance. The standard treatment for OC includes preliminary debulking surgery followed by platinum-based (carboplatin and cisplatin) and/or taxane family-based (paclitaxel and docetaxel) chemotherapy [164,165]. The mode of action of platinum-based therapies is oxidative stress-induced cellular damage and initiation of cell death pathways, such as apoptosis, which is triggered by this class of chemotherapeutics [166]. The taxane family-based drugs are used to inhibit cell division through microtubule stabilization [167]. However, due to toxic side effects associated with high-dose treatment and acquired resistance to carboplatin and cisplatin treatment, this traditional route of therapy has critical limitations. The mechanisms of drug resistance to these treatment options include an increase in DNA damage repair and an increase in antioxidant production to detoxify cancer cells [168]. Regarding this, other chemotherapeutic agents are used to treat resistant tumors, including gemcitabine, doxorubicin, and bevacizumab [169,170,171]. There are additional combinations of chemotherapy used to treat OC, such as targeted treatment of anti-apoptotic proteins that are overexpressed in OC cells. For instance, the anti-apoptotic protein Bcl-2 is overexpressed in OC, and treatment of tumor cells with a combination of cisplatin or carboplatin and Bcl-2 inhibitors show an increased level of cancer cell death induction [172,173,174,175]. The class of VEGF regulators known as specific proteins (Sp) are also targeted by small molecule inhibitors to induce cell death [176]. Another important mechanism for cancer cells to evade treatment is the upregulation of the glycoproteins that form the molecular pumps to export chemotherapeutic agents out of the cancer cells, driving the process of multi-drug resistance [177,178]. The complexity of miRNA dysregulation in OC also contributes to treatment resistance. For instance, OC cells evade apoptosis in response to paclitaxel treatment through upregulation of miR-21 and miR-106a, that target and downregulate the pro-apoptotic proteins APAF1 and CASP7, respectively [131,132]. Similarly, miR-182 upregulation in OC results in evasion of apoptosis in response to cisplatin/paclitaxel treatment through the downregulation of pro-apoptotic protein PDCD4 [142]. Regarding therapeutic efficacy, the ROS-induced miRNAs mentioned previously, miR-200a and miR-141, although shown as oncogenic, can increase the sensitivity of OC to paclitaxel treatment through the downregulation of p38 [121,179]. Similarly, overexpression of miR-522 can increase the sensitivity of OC to paclitaxel treatment [180]. A better understanding of the molecular mechanisms driving treatment resistance in OC is of vital importance for the design of therapies that will effectively treat aggressive, resistant tumors.

The increase in intracellular ROS levels in OC has been shown to contribute to therapeutic resistance. For instance, an increase in ROS in OC results in the overexpression of dCTP pyrophosphatase I (DCTPP1), which has a role in DNA damage repair and plays a major contribution to cisplatin resistance [181]. By a differing mechanism, the upregulation of calcium/calmodulin-dependent protein kinase II gamma (CAMK2G) in response to increasing levels of ROS reprograms the cellular redox system through the phosphorylation of inositol triphosphate3-kinase B (ITPKB), resulting in adaptive redox homeostasis and increased resistance to cisplatin treatment [182]. Similarly, the upregulation of PGC1-α by increasing intracellular ROS contributes to chemotherapy resistance through the upregulation of drug resistance-related proteins, MDR1 and ABCG2, leading to increased antioxidant production and drug efflux [183]. The increase in ROS in OC downregulates miR-199a and miR-125b, resulting in the increased expression of HER2 and HER3 and therapeutic resistance [114]. Thus, another mode of treatment for OC is vaccines targeting human HER2 and HER3 [184,185]. In work highlighted here, the increase in NOX4-derived ROS contributes to therapeutic resistance in OC through the upregulation of HER3. The upregulation of HER3 is a clinical marker for OC, which is positively correlated with poor prognosis [186]. NOX4 directly activates HER3 and contributes to the increased resistance of OC cells to chemotherapy and radiation treatments [113]. The deletion of *NOX4* results in a reduction in the therapeutic resistance of OC cells [113]. Similarly, inhibition of NOX4 acts synergistically with HER3 inhibition to decrease tumor growth in OC [113]. The knockdown of NOX4 using siRNA also results in enhanced sensitivity to radiation treatment in OC cells, proving this pathway relevant in multi-modal therapeutic resistance [113]. The NOX4-driven system of endogenous ROS production demonstrates a new mechanism in OC cells to promote tumor development, angiogenesis, and an increase in therapeutic resistance through the upregulation of HER3, reflecting a candidate for targeted therapy of treatment-resistant OC (Figure 2). These findings shed light on the importance of endogenous NOX4-derived ROS production in cell signaling and the progression of OC and the propensity of tumors to evade current lines of treatment.

## 7. Future Directions

The key to effective treatment of OC is the understanding of the molecular mechanisms that drive tumor development and resistance to current treatments. In the system described above, the increased levels of endogenous ROS produced by NOX4 is utilized by OC cells to stimulate tumorigenesis, angiogenesis, and treatment resistance (Figure 2). This adaptation in cellular signaling allows OC tumors to proliferate and develop resistance to chemotherapeutics through ROS production and upregulation of HER3, thus identifying this NOX4-driven pathway as a potential target for the treatment of chemoresistant tumors. In support of this, clinical trial studies show HER3 upregulation is associated with poor prognosis in OC, which serves as a clinical marker of tumor development, and HER3 expression is induced in response to current chemotherapeutics agents [186,187]. Therefore, this pathway provides an explanation for the ineffectiveness of traditional therapies for advanced OC and the development of therapeutic resistance. The implications of the findings reviewed here include the potential for NOX4 overexpression and increased levels of ROS to be utilized as a diagnostic biomarker in OC. Furthermore, there is clinical relevance for identifying new treatable targets in OC affected by this NOX4-driven system, particularly in resistant tumors.

As a significant mediator of miRNA dysregulation, ROS can have widespread effects on cellular processes. The roles of miRNA dysregulation in OC complicate the understanding of signaling pathways altered by tumors, with some acting as oncogenes and others acting as tumor suppressors. Similarly, the dysregulation of miRNAs in a cell-type-specific manner provides an opportunity to target specific miRNAs in different types of cancers. This could be accomplished by targeting the suppression of oncogenic miRNAs, which are typically upregulated in tumors, whereas the expression levels of tumor suppressor-like miRNAs are typically downregulated or lost in tumors [188]. The suppression of oncogenic miRNAs can be achieved with the use of anti-miRNA molecules targeting specific miRNA for inhibition or degradation [189]. For instance, anti-miR-21 treatment in breast cancer and glioblastoma induces apoptosis through the inhibition of PI3K signaling [190,191]. Alternatively, the upregulation of tumor suppressor miRNAs can be achieved with the use of miRNA mimics, which are delivered as mature miRNA molecules [192]. The use of miRNA mimics in combination with other forms of therapies improves treatment efficacy and the elimination of tumor cells. For example, the delivery of miR-204-5p in combination with oxaliplatin in colon cancer reduced tumor growth and induced apoptosis [193]. In further support of this, the treatment of relapsed, multidrug-resistant OC tumors with anti-Let-7 improved the efficacy of paclitaxel-induced cell death [194]. Since miRNAs are upstream regulators of a variety of cellular processes, the manipulation of their expression could cause adverse effects on surrounding tissues [195]. However, current research focusing on miRNA dysregulation is deciphering the mechanisms by which miRNAs affect different types of cancer. The increasing understanding of miRNA dysregulations in OC will allow for more direct targeting of the molecular pathways that are altered at each stage of tumor development. In addition, the up or downregulation of certain miRNAs in OC can also act as diagnostic biomarkers, as they have been demonstrated to have potential in many different cancer types [196]. Altogether, the altered molecular mechanisms driving OC development and treatment resistance are in part regulated by increased levels of endogenous ROS production and miRNA dysregulations. There are potentially new opportunities for more effective treatment of advanced OC by targeting the overlap in signaling pathways between these two mechanisms. However, limitations in our complete understanding of the roles of increased ROS and miRNA dysregulations in OC development necessitate more research efforts in these areas of study.

## Figures and Tables

**Figure 1 ijms-23-06702-f001:**
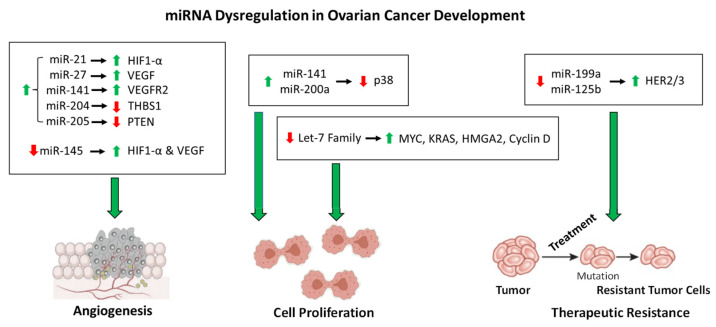
**The roles of miRNA dysfunction in OC development.** As described in the text, the miRNAs shown above are dysregulated in OC that contribute to angiogenesis, cell proliferation, and therapeutic resistance. The dysregulation of each miRNA is denoted by an up arrow (upregulation) or a down arrow(downregulation). The regulation of proteins affected by the dysregulation miRNAs is denoted in the same manner.

**Figure 2 ijms-23-06702-f002:**
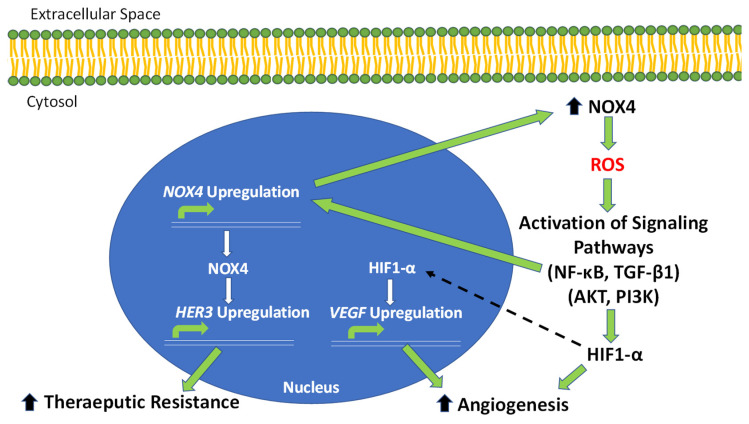
**NOX4-driven pathways of OC tumor progression, angiogenesis and therapeutic resistance.** The overexpression of NOX4 in OC results in an increase in intracellular ROS production. Increased ROS leads to an increase in HIF1-A through activating PI3K and AKT signaling. HIF1-α then activates the critical angiogenic factor, VEGF. Increased ROS also activates NF-κB and TGF-β1 signaling, which lead to the direct upregulation of NOX4. The increase in NOX4 contributes, in a positive-feedback manner, to increased ROS production. NOX4 also activates the expression of HER3, contributing to therapeutic resistance in OC tumors.

**Table 1 ijms-23-06702-t001:** miRNAs upregulated in OC cells that contribute to tumor growth and development, angiogenesis, and therapeutic resistance.

miRNA	Target mRNA(s)/Function	Ref
miR-21	*APAF1*/Promotes angiogenesis and treatment resistance (paclitaxel)	[131,132]
miR-22	*MXI1*/Promotes tumor growth	[133]
miR-27a	*VEGF*/Promotes *VEGF* expression to promote angiogenesis	[123]
miR-30a	*FOXD1*/Promotes cell cycle progression and growth	[134]
miR-92a	*DKK1*/Promotes Wnt signaling and tumor growth	[135]
miR-99a	*FN1, VTN*/Promotes tumor metastasis	[136]
miR-106a	*CASP7*/Promotes treatment resistance (paclitaxel)	[137]
miR-141	*p38*α, *KEAP1*/Promotes tumor growth and treatment resistance (cisplatin)	[121,138]
miR-181a	*SMAD7*/Promotes tumor growth, angiogenesis, and treatment resistance	[139]
miR-182	*MTSS1, PDCD4*/Promotes tumor growth, metastasis, and treatment resistance (cisplatin and paclitaxel)	[140,141,142]
miR-200a	*p38*α/Promotes tumor growth	[121]
miR-203	*PDHB*/Promotes cell proliferation	[143]
miR-204	*THBS1*/Promotes angiogenesis and tumor growth	[126,127]
miR-205	*SMAD4, PTEN*/Promotes metastasis and angiogenesis	[144]
miR-210	*PTPN1*/Promotes survival and evasion of cell death mechanisms	[145]
miR-214	*PTEN*/Promote cell survival and treatment resistance (cisplatin)	[146]
miR-223	*PTEN*/Promote tumor growth and treatment resistance (cisplatin)	[147]
miR-376a	*KLF15*/Promotes cell cycle progression and growth	[148]
miR-443	*MAD2*/Promote tumor growth and treatment resistance (paclitaxel)	[149]
miR-551b	*STAT3*/Promotes tumor growth	[150]
miR-552	*PTEN*/Promotes tumor metastasis	[151]
miR-622	*KU70, KU80*/Promotes treatment resistance (cisplatin and PARP Inhibitors)	[152]
miR-939	*APC2*/Promotes Wnt signaling and tumor growth	[153]
miR-1246	*CAV1*/Promotes treatment resistance (paclitaxel)	[154]

**Table 2 ijms-23-06702-t002:** miRNAs downregulated in OC cells that function as tumor suppressors.

miRNA	Target mRNA(s)/Function	Ref
Let-7 Family	*KRAS, c-MYC*/Tumor suppression through *KRAS* and *c-MYC* downregulation.	[86,87]
miR-31	*CDKN2A*/Tumor suppression through *CDKN2A* downregulation.	[155]
miR-125b	*VEGF, HER3, HIF1-α*/Tumor suppression through *VEGF*, *HER3* and *HIF1-α* downregulation.	[114]
miR-135a	*CCR2*/Tumor suppression through *CCR2* degradation.	[156]
miR-145	*P70S6K1*/Tumor suppression through *P70S6K1* downregulation.	[129]
miR-181	*RTKN2*/Tumor suppression through *RTKN2* downregulation.	[157]
miR-199a	*HER3*/Tumor suppression through *HER3* downregulation.	[114]
miR-200b/200c	*DNMT3A/3B*/Tumor suppression through *DNMT3A/3B* downregulation.	[158]
miR-206	*c-MET*/Tumor suppression through *c-MET* downregulation.	[159]
miR-298	*EZH2*/Tumor suppression through *EZH2* downregulation.	[160]
miR-424	*CCNE1*/Tumor suppression through *CCNE1* downregulation.	[161]
miR-490	*CDK1*/Tumor suppression through *CDK1* downregulation.	[162]
miR-508	*MAPK1*/Tumor suppression through *MAPK1* downregulation.	[163]

## Data Availability

The data presented in this study are openly available in Medline and Embase.

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
