# Peer review of "ROS and miRNA Dysregulation in Ovarian Cancer Development, Angiogenesis and Therapeutic Resistance"

_ijms, 2022, doi:10.3390/ijms23126702_

Round 1
Reviewer 1 Report
Dear Editor,
The authors of the above manuscript have reviewed the recent researches of both ROS and miRNA dysregulation in ovarian cancer highlighting the contribution of miRNAs, NOX4 and endogenous ROS production in tumor progression, angiogenesis and resistance to chemotherapies. Overall, the review is well constructed and I have few suggestions before acceptance for publication.
Abstract: avoid repetitions and I would reformulate the last sentence from lane 19 to 22
Lane 113 and 114 : I would link the two phrases together
Lane 127: a reference is missing
Lane 152: I would expand a bit more the role of NOX4 in ROS production in the context of OC especially as potential target for therapies.
Paragraph 4: I suggest the authors to explain a bit more how the mRNA expression profile changes in different OC subtypes
Lane 196: I would expand a bit the contribution of the most studied mRNAs in cancer progression and add appropriate references
Lane 221: I would avoid repetitions (dysregulation) and maybe link the two sentences and add reference at lane 224
Paragraph 5: I would also mention the importance of circulatory miRNAs expression to predict disease outcome or as biomarkers for OC and I suggest the authors to make another table with the miRNAs that exert a tumor suppressor role.
Figure 1: the arrows in the figure legend are not visible. Also, it would be nice to add a picture underneath “Cell proliferation” and Therapeutic resistance” as it has been done for “Angiogenesis”.
Paragraph 6: I would expand the importance of miRNAs in chemotherapy resistance and I suggest the authors to mention/describe in the text the role of miRNAs 200a and 141 in paclitaxel sensitivity as it occurs in a ROS dependent manner (M.Bogdan et al. 2011 Nature Medicine; Liu N. et al. 2015 Neoplasma) and also miRNA 522 in paclitaxel resistance OC (Myamoto M. et al. Scientific Reports 2020).
Author Response
Abstract: avoid repetitions and I would reformulate the last sentence from lane 19 to 22
The response: This correction has been made.
Lane 113 and 114 : I would link the two phrases together
The response: This correction has been made.
Lane 127: a reference is missing
The response: We add the reference.
Lane 152: I would expand a bit more the role of NOX4 in ROS production in the context of OC especially as potential target for therapies.
The response: The concluding statement of the paragraph has been changed to reflect this suggestion.
Paragraph 4: I suggest the authors to explain a bit more how the mRNA expression profile changes in different OC subtypes
The response: A short statement with the appropriate reference has been added in the revised version.
Lane 196: I would expand a bit the contribution of the most studied mRNAs in cancer progression and add appropriate references
The response: This has been addressed by adding the description to the paragraph.
Lane 221: I would avoid repetitions (dysregulation) and maybe link the two sentences and add reference at lane 224
The response: These corrections have been made.
Paragraph 5: I would also mention the importance of circulatory miRNAs expression to predict disease outcome or as biomarkers for OC and I suggest the authors to make another table with the miRNAs that exert a tumor suppressor role.
The response: A statement regarding circulatory miRNAs was added. A table showing common miRNAs that are downregulated in OC was added with the reference.
Figure 1: the arrows in the figure legend are not visible. Also, it would be nice to add a picture underneath “Cell proliferation” and Therapeutic resistance” as it has been done for “Angiogenesis”.
The response: These changes have been made.
Paragraph 6: I would expand the importance of miRNAs in chemotherapy resistance and I suggest the authors to mention/describe in the text the role of miRNAs 200a and 141 in paclitaxel sensitivity as it occurs in a ROS dependent manner (M.Bogdan et al. 2011 Nature Medicine; Liu N. et al. 2015 Neoplasma) and also miRNA 522 in paclitaxel resistance OC (Myamoto M. et al. Scientific Reports 2020).
The response: This new description and reference have been added in the revised manuscript.
Reviewer 2 Report
The Authors present a Review entitled "ROS and miRNA Dysregulation in Ovarian Cancer Development, Angiogenesis and Therapeutic Resistance" in which they focus on the discussion of several important miRNAs, NOX4, and ROS that contribute to therapeutic resistance in ovarian cancer, highlighting the potential for the effective treatment of OC through the targeting of certain miRNAs, NOX4 and endogenous ROS. Overall, the review is well written with a comprehensive background and appropriate details.
Minor:
I would suggest a spell check for typos.
Major:
I suggest the author to add a graphical scheme (in the form of a new figure) to better clarify in Paragraph 6 the potential mechanism of ROS in therapeutic resistance in ovarian cancer that the Authors propose.
Author Response
Minor:
I would suggest a spell check for typos.
Major:
I suggest the author to add a graphical scheme (in the form of a new figure) to better clarify in Paragraph 6 the potential mechanism of ROS in therapeutic resistance in ovarian cancer that the Authors propose.
The response: We performed a spell check to correct the typos. A new figure (Fig. 2) has been added to illustrate the regulation mechanism. The previous Figure 2 has been deleted to avoid redundancy.
Reviewer 3 Report
The review is interesting and well written.
References need to be conformed to journal style
Author Response
References need to be conformed to journal style
The response: References have been formatted accordingly to the journal style.
Round 2
Reviewer 2 Report
The authors have replied to all the comments